# Factors contributing to neonatal mortality in a tertiary center in south West Bank: A single-center retrospective cohort study

Ahmad Abu Sharkh[1], Farah Ighneimat[2], Yazan F. Khdour[3]*, Noor Y. Aladam[2], Inad Nawajaa[4], Alaa Al Janazerah[1], Insaf Najajrah[1], Beesan Maraqa[2]

**1** Department of Pediatrics, PRCS Hospital, Hebron, Palestine, **2** College of Medicine, Hebron University, Hebron, Palestine, **3** Department of Internal Medicine, Beit Jala Hospital, Bethlehem, Palestine, **4** College of Science and Technology, Hebron University, Hebron, Palestine

\* yazan.khdour97@gmail.com

## Abstract

### Background

Neonatal mortality remains a critical public health issue, particularly in low- and middle-income countries, where factors such as preterm birth, low birth weight, congenital anomalies, infections, and limited access to quality healthcare contribute significantly to neonatal deaths. This study examines neonatal mortality outcomes in the Neonatal Intensive Care Unit (NICU) at the Palestine Red Crescent Society (PRCS) Hospital in Hebron, Palestine for newborns that was transferred to PRCS from other hospitals. An institutional-based cohort study was conducted on 606 neonates admitted "transferred from other hospitals" to the NICU at PRCS Hospital from 2019 to 2024. Data were collected from the hospital's electronic registry, capturing clinical parameters and potential risk factors. Statistical analysis, including bivariate and multivariable logistic regressions, was performed using SPSS version 25 to evaluate neonatal mortality risk variables. Of the 606 neonates admitted "transferred from other hospitals" to the NICU over five years, 21.5% died, reflecting a substantial neonatal mortality rate. The study identified significant associations between neonatal mortality and sepsis (p = 0.001, aOR=2.34), intraventricular hemorrhage (p < 0.001, aOR=4.67), and necrotizing enterocolitis (p = 0.001, aOR=3.58). Transfer process, Sepsis, intraventricular hemorrhage, necrotizing enterocolitis, prematurity, low birth weight, and hypothermia were key factors associated with neonatal mortality in this NICU setting. Prioritizing early management of sepsis, NEC, and low birth weight is crucial to reducing neonatal deaths in institutional settings. These findings can guide interventions to improve neonatal outcomes and support healthcare facilities in transfer process training for high-risk newborns to reduce preventable deaths.

**Data availability statement:** We confirm that all data underlying the findings in our manuscript are held at the Palestine Red Crescent Hospital. Due to institutional policies and patient confidentiality, the full de-identified data set cannot be publicly shared at this time. However, interested researchers may request access to the data through the NICU department. Please direct any data access inquiries to: dr-abu-mayaleh@hotmail.com Medical director Dr Abdul Razzaq Abu Mayaleh

**Funding:** The author(s) received no specific funding for this work.

**Competing interests:** The authors have declared that no competing interests exist.

## Introduction

Neonatal mortality (NM), defined as the death of a newborn within the first 28 days of life, remains a critical public health concern globally [1]. While significant strides have been made in reducing child mortality in recent decades, neonatal mortality rates continue to pose challenges in many parts of the world [2].

Preterm birth (defined as birth before 37 weeks of pregnancy) is the most common cause of neonatal mortality (NM). Babies born before 37 weeks of gestation may have more health complications than full-term infants, including those with congenital anomalies or birth defects [3]. Low birth weight (less than 2,500 grams [5.5 pounds]) is another significant risk factor for NM [4]. Additionally, congenital anomalies themselves are a contributing risk to neonatal mortality [5].

Low-and middle-income countries account for the vast majority of maternal, fetal and neonatal deaths in the world. Comprehensive analyses of maternal, neonatal, and fetal deaths have shown that quality obstetric care provided during delivery, including access to cesarean delivery, could significantly reduce such deaths [6]. A study conducted in southern Ethiopia showed that the neonatal mortality incidence was 27 per 1000 neonates. Predictors of neonatal mortality were: multiple births, mothers who did not attend antenatal care visits, neonates born by cesarean section, not initiated breastfeeding within one h of birth, neonates resuscitated, hyaline membrane disease, and perinatal asphyxia [7]. Another study conducted in India showed that the major cause of admission was perinatal asphyxia followed by prematurity. Sepsis was the major cause of mortality and a strong association was found between mortality and maturity status [8].

Neonatal mortality is a complex outcome influenced by a myriad of biological, socio-economic, and environmental factors.Transfer procees Premature birth, low birth weight, congenital anomalies, and maternal health conditions are among the key determinants that can impact a newborn's chances of survival. Mortality rates in NICUs vary between countries; however, no published studies have explored NICU mortality rates in Palestine. Our study aimed to determine the NICU mortality rate in specific group of NICU admission, gather causes, and detect the major ones. Understanding the factors contributing to neonatal mortality is essential for developing effective strategies to prevent and address this issue.

## Methods

### Statement of ethics

The College of Medicine Institutional Review Board reviewed and approved this study protocol, approval number *Ref: ER.CM.23.2024.*

### Study design and sample

A retrospective cohort study analyzed neonatal outcomes in the Neonatal Intensive Care Unit (NICU) at the Palestinian Red Crescent Society (PRCS). The study included all neonates (Day 1–28) admitted "transferred" to the NICU at PRCS between January 1, 2019, and December 31, 2024.

## Data collection

Data were extracted from medical records, NICU logs, and hospital databases using a standardized collection, which is Electronic Health Records (EHRs), & Databases, Data were accessed on January 1, 2025. Variables were categorized into maternal and neonatal characteristics. Maternal data included age, pregnancy complications (e.g., pregnancy-induced hypertension, preterm rupture of membranes), delivery complications, and delivery site and mode. Neonatal data included gestational age, birth weight, birth type (singleton or multiple), referred status, admission temperature, and the primary cause of NICU admission (e.g., congenital anomalies, intraventricular hemorrhage, asphyxia, sepsis, necrotizing enterocolitis, pneumothorax, respiratory distress syndrome, or meconium aspiration syndrome). The primary outcome of interest was neonatal mortality. Data were verified for completeness and consistency before analysis, ensuring the reliability of the findings. Ethical approval was obtained from the Institutional Review Board at the College of Medicine in Hebron University IRB number; ER.CM.23.2024 and all data were anonymized to maintain confidentiality. We obtained formal approval from the PCRS as the data was sourced from preexisting records, hence written informed consent was not required.

## Data analysis

Data were aggregated in Microsoft Excel and analyzed using SPSS version (25). Statistical analyses were conducted in two stages. First, chi-square tests evaluated associations between neonatal mortality and categorical predictors (e.g., maternal preeclampsia, preterm birth). Second, variables meeting significance thresholds ($p < 0.05$) or clinical relevance were included in a binary logistic regression model to identify independent predictors. Adjusted odds ratios (aOR) were calculated to account for confounders such as gestational age and birth weight.

## Results

The study included 606 neonates admitted transferred to the NICU over 5 years. Positive outcomes (survival rate) were observed in 78.5% of cases, 57.3% being normal vaginal deliveries and 42.7% cesarean sections. About 79% of admissions were on the first day of life, 13.9% between 1–6 days, and 7.1% after 6 days. Male neonates accounted for 58.7% of admissions. Nearly half (49.3%) were full-term (>37 weeks), 33.8% preterm (32–37 weeks), 9.9% very preterm (28–32 weeks), and 6.9% extremely preterm (<28 weeks). Over half (55.1%) stayed >7 days, 27.7% stayed 1–3 days, and 17.2% stayed 4–6 days, indicating the severity of conditions requiring extended care, as shown in Table 1.

All neonates had hypothermia most of them (69.1%) had mild hypothermia (36–36.4°C), (rectally) while 30.4% had moderate hypothermia (32–36°C), and only 0.5% had severe hypothermia (<32°C). Congenital abnormalities were observed in 20.5%. Birth asphyxia occurred in 4%, and meconium aspiration in 3.3%. Sepsis affected 30.2%, indicating a significant infection burden. The majority did not experience NEC (95.4%), pneumothorax (94.1%), RDS (66.5%), IVH (95.5%), PROM (91.7%), or PPROM (97.5%). Additionally, 34.7% underwent detailed ultrasounds, as shown in Table 2.

Most mothers (92.9%) were reported as healthy (gleaned from the electronic health record), while 7.1% had some health issue. Respectively, preeclampsia, GDM, infectious disease, and PPHN were present in (3%), (2%), (8.3%), and (6.8%) of cases. As shown in Table 3.

The primary outcome of our study revealed that approximately 22% of neonates died, while 78% recovered.

Neonates aged 1–6 days demonstrated the highest likelihood of recovery, with an odds ratio (OR) of 4.98 compared to older age groups. In contrast, lower birth weights were associated with progressively reduced odds of recovery, though these findings were not statistically significant. For infants weighing less than 1 kg, the adjusted odds ratio (aOR) was 0.34 (95% CI: 0.05–2.43; p 0.28), suggesting no meaningful association with recovery. Similarly, the 1–1.5 kg weight group had an aOR of 0.43 (95% CI: 0.08–2.23; p 0.317), further supporting the lack of statistical significance. The trend continued in the 1.5–2.5 kg category, where the aOR was 0.49 (95% CI: 0.17–1.44; $p = 0.195$), indicating a potential but non-significant decrease in recovery odds compared to weightier infants. Reference group members weighed above 2.5 kg. No weight category was statistically associated with mortality, although lower birth weight was associated with higher mortality

**Table 1. Sociodemographic characteristics of the study participants (n = 606).**

| Variable | Sub Variable | n | % |
|---|---|---|---|
| Mode of delivery | NSVD | 347 | 57.3 |
| | CS | 259 | 42.7 |
| Age of admission | <1 day | 479 | 79 |
| | 1–6 day | 84 | 13.9 |
| | >6 = 3 day | 43 | 7.1 |
| Gender | Female | 250 | 41.3 |
| | Male | 356 | 58.7 |
| Maturity | <28 week | 42 | 6.9 |
| | 28–32 week | 60 | 9.9 |
| | 32–37 week | 205 | 33.8 |
| | >37 week | 299 | 49.3 |
| Weight | <1 kg | 43 | 7.1 |
| | 1–1.5 kg | 53 | 8.7 |
| | 1.5–2.5 kg | 192 | 31.7 |
| | >2.5 kg | 318 | 52.5 |
| Duration of stay | 1–3 day | 168 | 27.7 |
| | 4–6 day | 104 | 17.2 |
| | >7 day | 334 | 55.1 |

NSVD: normal spontaneous vaginal delivery; CS: cesarean section

risk. Shorter hospital stays were associated with higher mortality risk, with an adjusted hazard ratio of 0.07 (95% CI: 0.03-0.17). People with a mild hypothermia 36-36.4°C have the lowest mortality rate (15.8%) and the best recovery rate (84.2%) as seen in Table 4.

Sepsis increased mortality risk (aOR: 2.34, 95% CI: 1.17–4.71, p = 0.017). NEC showed a trend toward higher mortality (aOR: 3.58, 95% CI: 0.92–13.9, p = 0.065), though not statistically significant. Pneumothorax (aOR: 2.05, 95% CI: 0.70–5.95, p = 0.189) had no strong association with mortality. RDS was strongly associated with mortality (aOR: 8.08, 95% CI: 3.81–17.14, p = 0.001), as was IVH (aOR: 4.67, 95% CI: 1.06–20.54, p = 0.041). Lack of detailed ultrasound increased mortality risk (aOR: 2.76, 95% CI: 1.36–5.59, p = 0.005). PROM (aOR: 1.03, 95% CI: 0.37–2.85, p = 0.949) showed no significant association with mortality, as detailed in Table 5.

## Discussion

This study focused on determining the incidence of mortality and identifying different factors affecting neonatal mortality among a specific group of neonates that was transferred to PRCS from other hospitals between 2019 and 2024 in the neonatal intensive care unit of the Palestine Red Crescent Society Hospital. 21.5% of the neonates died, highlighting a substantial neonatal mortality rate for this specific group. Palestine Red Crescent Society Hospital considered as highly qualified tertiary hospital, most of transferred newborns was critical or had congenital abnormality or moderate to extreme premature so was transferred to tertiary center and all were subjected to transfer process also all cases was hypothermic on admission " all these factors contribute and explain the high mortality rate in this specific group, highlights the urgency need for development of transport process. during the same period 2019–2024 admissions to the same NICU that was delivered at the same hospital was 2120 cases with 86 cases dies "mortality rate was 4.05%. the incidence of mortality in other Arab countries, such as Qatar, where the mortality rate is 6.5% [9]. Also, in Sidama regional state, Ethiopia (14.2%), which is considered a low-income country [10].

**Table 2. The most common conditions and complications that affect the study participant (n = 606).**

| Variable | Sub Variable | n | % |
|---|---|---|---|
| Temperature | 36–36.4 | 419 | 69.1 |
|  | 32–36 | 184 | 30.4 |
|  | <32 | 3 | 0.5 |
| Congenital abnormality | No | 482 | 79.5 |
|  | Yes | 124 | 20.5 |
| Birth asphyxia | No | 582 | 96 |
|  | Yes | 24 | 4 |
| Meconium aspiration | No | 586 | 96.7 |
|  | Yes | 20 | 3.3 |
| Sepsis | No | 423 | 69.8 |
|  | Yes | 183 | 30.2 |
| Necrotizing enterocolitis (NEC) | No | 578 | 95.4 |
|  | Yes | 28 | 4.6 |
| Pneumothorax | No | 570 | 94.1 |
|  | Yes | 36 | 5.9 |
| Respiratory distress syndrome (RDS) | No | 403 | 66.5 |
|  | Yes | 203 | 33.5 |
| Intraventricular hemorrhage (IVH) | No | 579 | 95.5 |
|  | Yes | 27 | 4.5 |
| Detailed ultrasound | No | 396 | 65.3 |
|  | Yes | 210 | 34.7 |
| Premature rupture of membranes (PROM) | No | 556 | 91.7 |
|  | Yes | 50 | 8.3 |
| preterm premature rupture of membranes (PPROM) | No | 591 | 97.5 |
|  | Yes | 15 | 2.5 |
| Maternal age | (Min, Max); Mean±SD | (16,47); 27.41 ± 6.09 | |

Most deliveries were normal vaginal deliveries (NVD), accounting for 57.3% of cases, while 42.7% were cesarean sections (CS). The nearly equal distribution highlights a relatively high rate of cesarean sections, suggesting that a considerable number of births required surgical intervention, which may be due to complications during labor or preexisting conditions. The country's cesarean section rate was higher than the global average [11]. For nearly 30 years, the international healthcare community has considered the ideal rate for CS delivery to be between 10% and 15%. Still, its rates have risen from around 7% in 1990 to 21% today, according to WHO [12]. In comparison with vaginal birth, neonates born after CS delivery have significantly higher rates of respiratory morbidity and NICU admission and a longer length of hospital stay [13].

The study found a statistically significant association between sepsis, IVH, NEC, and neonatal mortality. Among neonates with sepsis, 45.4% died compared to 54.6% who recovered, with the impact on mortality being lower than reported in other studies. IVH was strongly associated with mortality, with a death rate of 74.1% in affected patients compared to 19% in those without, consistent with findings from other countries [14]. NEC was also significantly linked to increased mortality, with a death rate of 57.1% among affected neonates, supporting prior research identifying NEC as a leading cause of morbidity and mortality in preterm infants [15]. RDS is a significant predictor of mortality, with 51.2% of affected neonates dying compared to 6.5% of those without RDS, demonstrating a strong association with poor outcomes. The condition, driven by surfactant deficiency and immature lung development, leads to alveolar collapse, poor gas exchange,

**Table 3. Other conditions affect the study participants (n = 606).**

| Variable | Sub variable | Frequency | Percentage |
|---|---|---|---|
| MgSO4 | No | 597 | 98.5 |
| | Yes | 9 | 1.5 |
| Mother health | Healthy | 563 | 92.9 |
| | Unhealthy | 43 | 7.1 |
| Preeclampsia | No | 588 | 97 |
| | Yes | 18 | 3 |
| Gestational diabetes mellitus (GDM) | No | 594 | 98 |
| | Yes | 12 | 2 |
| Infectious disease | No | 556 | 91.7 |
| | Yes | 50 | 8.3 |
| Maternal age | (Min, Max); Mean±SD | (16,47); 27.41±6.09 | |
| Persistent pulmonary hypertension of the newborn | No | 565 | 93.2 |
| | Yes | 41 | 6.8 |
| weight | <1 kg | 43 | 7.1 |
| | 1–1.5 kg | 53 | 8.7 |
| | 1.5–2.5 kg | 192 | 31.7 |
| | >2.5 kg | 318 | 52.5 |

**Table 4. Factors related to delivery and admission and its association with the outcome.**

| Variables | | Outcome | | | Multivariate analysis | |
|---|---|---|---|---|---|---|
| | | Died | recovered | p-value | aOR (95%CI) | aP-value |
| Age of admission | <1 day | 117 (24.4%) | 362 (75.6%) | 0.001 | 1.4 (0.43–4.61) | 0.225 |
| | 1–6 day | 3 (3.6%) | 81% (96.4) | | 4.98 (0.75–33.27) | 0.58 |
| | >6 day | 10 (23.3%) | 33 (76.7%) | | 1 | 0.097 |
| Maturity | <28 week | 35 (82.3%) | 7 (16.7%) | 0.001 | 0.15 (0.02–1.14) | 0.066 |
| | 28–32 week | 31 (51.7%) | 29 (48.3%) | | 0.28 (0.06–1.32) | 0.108 |
| | 32–37 week | 27 (13.2%) | 178 (86.8%) | | 0.94 (0.32–2.80) | 0.915 |
| | >37 week | 37 (12.4%) | 262 (87.6%) | | 1 | |
| weight | <1 kg | 33 (76.7%) | 10 (23.3%) | 0.001 | 0.34 (0.05–2.43) | 0.28 |
| | 1–1.5 kg | 26 (49.1%) | 27 (50.9%) | | 0.43 (0.08–2.23) | 0.317 |
| | 1.5–2.5 kg | 36 (18.8%) | 156 (81.2%) | | 0.49 (0.17–1.44) | 0.195 |
| | >2.5 kg | 35 (11%) | 283 (89%) | | 1 | |
| Duration of stay | 1–3 day | 62 (36.9%) | 106 (63.1%) | 0.001 | 0.07 (0.03–0.17) | 0.001 |
| | 4–6 day | 15 (14.4%) | 89 (85.6%) | | 0.37 (0.12–1.18) | 0.055 |
| | >7 day | 53 (15.9%) | 281 (84.1%) | | 1 | |
| Temperature | 36–36.4 | 66 (15.8%) | 353 (84.2%) | 0.001 | 4.24 (0.28–64.98) | 0.30 |
| | 32–36 | 62 (33.7%) | 122 (66.3%) | | 4.06 (0.26–62.35) | 0.314 |
| | <32 | 2 (66.7%) | 1 (33.3%) | | | |

and hypoxemia, particularly in preterm infants. The prognosis of newborns has shown improvement and complications have decreased with oxygen therapy and continuous positive airway pressure (CPAP) and surfactant treatment [16].

Patients with PROM had a higher death rate (32%) compared to those without (20.5%). PROM is not statistically significant. However, this Suggesting a potential relationship worth further investigation. PROM is one of pregnancy

**Table 5. The most common conditions and complications affect the study participants and patient outcome.**

| Variable | Sub Variable | Died | Recovered | p-value | Multivariate analysis | |
|---|---|---|---|---|---|---|
| | | | | | aOR (95%CI) | aP-value |
| Congenital abnormality | No | 86 (17.8%) | 396 (82.2%) | 0.001 | 3.76 (1.76–8.07) | 0.001 |
| | Yes | 44 (36.5%) | 80 (64.5%) | | 1 | |
| Birth asphyxia | No | 117 (20.1%) | 465 (79.9%) | 0.001 | 11.05 (3.0–40.7) | 0.001 |
| | Yes | 13 (54.2%) | 11 (45.8%) | | 1 | |
| Meconium aspiration | No | 125 (21.3%) | 461 (78.7%) | 0.69 | 1.52 (0.31–7.41) | 0.606 |
| | Yes | 5 (25%) | 15 (75%) | | 1 | |
| Sepsis | No | 47 (11.1%) | 376 (88.9%) | 0.001 | 2.34 (1.17–4.71) | 0.017 |
| | Yes | 83 (45.4%) | 100 (54.6%) | | 1 | |
| Necrotizing enterocolitis (NEC) | No | 114 (19.7%) | 464 (80.3%) | 0.001 | 3.58 (0.92–13.9) | 0.065 |
| | Yes | 16 (57.1%) | 12 (42.9%) | | 1 | |
| Pneumothorax | No | 111 (19.5%) | 459 (80.5%) | 0.001 | 2.05 (0.70–5.95) | 0.189 |
| | Yes | 19 (52.8%) | 17 (47.2%) | | 1 | |
| Respiratory distress syndrome (RDS) | No | 26 (6.5%) | 377 (93.5%) | 0.001 | 8.08 (3.81–17.14) | 0.001 |
| | Yes | 104 (51.2%) | 99 (48.8%) | | 1 | |
| Intraventricular haemorrhage (IVH) | No | 110 (19%) | 469 (81%) | 0.001 | 4.67 (1.06–20.54) | 0.041 |
| | Yes | 20 (74.1%) | 7 (25.9%) | | 1 | |
| Detailed ultrasound | No | 67 (16.9%) | 329 (83.1%) | 0.001 | 2.76 (1.36–5.59) | 0.005 |
| | Yes | 63 (30%) | 147 (70%) | | 1 | |
| Premature rupture of membranes (PROM) | No | 114 (20.5%) | 442 (79.5%) | 0.058 | 1.03 (0.37–2.85) | 0.949 |
| | Yes | 16 (32%) | 34 (68%) | | 1 | |
| preterm premature rupture of membrane | No | 121 (20.5%) | 470 (79.5%) | 0.001 | 1.28 (0.14–12.37) | 0.827 |
| | Yes | 9 (60%) | 6 (40%) | | 1 | |

complications which turn it to high risk pregnancy that needs argent work-up to the baby at the delivery room [17]. This goes with other studies, such as in Ethiopia, which found a strong association between PROM mothers and bad-health babies [18].

Temperature is another significant factor. As temperature decreases, the mortality rate increases, with extremely high death rates observed at temperatures below 32°C (66.7%). Hypothermia is one of the determinants of neonatal mortality [19]. Neonates are highly susceptible to temperature-related issues due to their immature thermoregulatory systems, limited fat stores, and high body surface area-to-weight ratio [20]. On the other hand, there is a limited ability to generate heat through shivering or brown fat metabolism. This affects mortality through metabolic disturbances; hypothermia increases glucose consumption, leading to hypoglycemia and metabolic acidosis. Also, it causes immune suppression and increases susceptibility to infections, including neonatal sepsis.

This study has several limitations. As a retrospective study, it relies on existing medical records, which may be incomplete or inaccurate, introducing potential bias. Conducted in a single NICU, the findings may not generalize to other institutions with different patient populations, practices, or resources. Neonates who were admitted to the NICU for less than 24 hours and had no identifiable risk factors were excluded from the analysis. These cases typically involved infants admitted for short-term observation and subsequently discharged by the attending physician after evaluation, as they were not deemed to require intensive care. Additionally, the study does not account for confounding factors such as socioeconomic status, parental health behaviors, or prenatal care quality, which could influence neonatal outcomes. These limitations should be considered when interpreting the results.

The problems participants experienced with prenatal care delays and unanticipated home deliveries and facility access challenges exist within the framework of Israel's occupation structural violence. The occupation's systemic restrictions, which include checkpoints, permit regimes, and land confiscation, have split the healthcare infrastructure while limiting freedom of movement, a fundamental health determinant [21]. The policies implemented by the government directly increase maternal risks because Leone et al. [22] discovered that increased conflict intensity leads to decreased maternal care utilization in the Occupied Palestinian Territory (OPT). The participants described how military operations and extended checkpoint delays interrupted their care, while economic instability and mental health issues made it harder for them to seek medical help. The solution to these inequities demands the elimination of structural violence through the removal of movement barriers and the establishment of decentralized healthcare systems to counteract the fragmentation caused by the occupation [21,22].

In conclusion, sepsis, IVH, NEC, prematurity, low birth weight, and hypothermia were significantly associated with neonatal mortality in the specialized neonatal care unit. Improving transfer process, Early management of sepsis and low birth weight should be the priority issues for controlling neonatal deaths in institutional settings. The results of this study can be utilized to prioritize and provide appropriate care to newborns who are at high risk for better prediction and to reduce preventable neonatal deaths in settings with limited resources.

## Acknowledgments

The authors express gratitude to the NICU staff and administration at PRCS Hospital for their support in providing access to essential data and resources for this study.

## Author contributions

**Conceptualization:** Alaa Al Janazerah.

**Data curation:** Farah Ighneimat, Noor Y. Aladam, Insaf Najajrah.

**Formal analysis:** Inad Nawajaa.

**Supervision:** Ahmad Abu Sharkh, Yazan Y. Khdour, Beesan Maraqa.

**Writing – original draft:** Farah Ighneimat, Noor Y. Aladam.

**Writing – review & editing:** Ahmad Abu Sharkh, Yazan Y. Khdour, Beesan Maraqa.

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
