## [Decision Letter · Decision Letter 0]

PGPH-D-25-00456

Factors contributing to neonatal mortality in a tertiary center in south West Bank: A Single-Center Retrospective Cohort Study

Dear Dr. khdour,

Thank you for submitting your manuscript to PLOS Global Public Health. After careful consideration, we feel that it has merit but does not fully meet PLOS Global Public Health’s publication criteria as it currently stands. Therefore, we invite you to submit a revised version of the manuscript that addresses the points raised during the review process.

In addition to the comments made my Reviewer #1 below, I believe the Discussion would be greatly strengthened by putting the study results in the context of current scholarship about the structural conditions faced by Palestinian women in the West Bank. There is strong evidence that the ongoing occupation by Israel has wide ranging impact on reproductive health. Please consider including a discussion of such conditions. I have pasted below two recent examples, but there are many other studies and papers on this topic.

- Asi YM. Freedom of Movement as a Determinant of Women’s Health: Global Analysis and Commentary. World Medical & Health Policy. 2021;13(4):641–52.

- Leone T, Alburez-Gutierrez D, Ghandour R, Coast E, Giacaman R. Maternal and child access to care and intensity of conflict in the occupied Palestinian territory: a pseudo-longitudinal analysis (2000–2014). Confl Health. 2019 Aug 7;13(1):36.

We look forward to receiving your revised manuscript.

Kind regards,

Sanghyuk S Shin

Academic Editor

Journal Requirements:

1. Please provide a complete Data Availability Statement in the submission form, ensuring you include all necessary access information or a reason for why you are unable to make your data freely accessible. If your research concerns only data provided within your submission, please write "All data are in the manuscript and/or supporting information files" as your Data Availability Statement.

2. Please insert an Ethics Statement at the beginning of your Methods section, under a subheading 'Ethics Statement'.

3. Tables should not be uploaded as individual files. Please remove these files and include the Tables in your manuscript file as editable, cell-based objects. For more information about how to format tables, see our guidelines: 

https://journals.plos.org/globalpublichealth/s/tables

4. We have noticed that you have cited Table 4 and 5 in the manuscript file but there are no corresponding tables in the manuscript. Please amend your manuscript to include this table, noting that tables should not be uploaded as individual files.

Additional Editor Comments (if provided):

Reviewers' comments:

Reviewer's Responses to Questions

**Comments to the Author**

1. Does this manuscript meet PLOS Global Public Health’s publication criteria?

Reviewer #1: Yes

2. Has the statistical analysis been performed appropriately and rigorously?

Reviewer #1: Yes

3. Have the authors made all data underlying the findings in their manuscript fully available (please refer to the Data Availability Statement at the start of the manuscript PDF file)?

Reviewer #1: No

4. Is the manuscript presented in an intelligible fashion and written in standard English?

Reviewer #1: Yes

Reviewer #1: There is no Table 4 or Table 5 to review. Please attach for proper review of entire manuscript and results. This should be a minor revision just to attach the tables.

Recommend copy editing for appropriate spelling, grammar, and use of abbreviates (e.g., lines 30-34).

Lines 48-49: this is a strong point and can speak to the lack of literature on Palestinian health in general. It may be worthwhile to mentioned studies that have focused on neonatal mortality in the region or in Palestinian refugee populations. For example, https://doi.org/10.1016/S0140-6736(13)60206-8 or https://doi.org/10.1016/S0140-6736(17)32061-5

Recommend rewording Line 54

Explicitly state any exclusion criteria in the study design and sample section, and the rationale. For example, why were neonates who were admitted to NICU for less than 24 hours not included.

Research method would be stronger if controls were included in the regression, like the factors listed in the Introduction section. However, since this is stated as a limitation, that should suffice but perhaps adding controls could be stated as something to explore as future research.

The methods used in the Data Analysis section were a little unclear. The use of chi-square tests and then multiple regressions could be more clearly stated with variables of interest clearly delineated. For example, why is preeclampsia and subsequent ESRD especially noted? Also, bivariate logistic regression was previously stated in the manuscript but not clearly defined (i.e., what are the two binary dependent variables?). Sometimes odds ratio is reported while other times adjusted odds ratio. Perhaps this would be clearer with Tables 4 and 5, but those were not attached to the manuscript.

Line 99: what qualifies a mother to be listed as “healthy”? Is this a self-report variable or gleaned from the electronic health record?

For Tables 1-3: there seems to be information from the mother and neonate dispersed throughout the tables. Consider organizing based on those two categories, where summary statistics on maternal characteristics are distinguished from patient admission characteristics.

Line 106: consider revising for more clear interpretation of the odds ratio.

Line 136: consider removing personal tense “our”.

Line 152-153: consider if mentioned if those therapies are available in Palestine, and the overall resource scarcity that may complicate treatment of neonates.

Line 154: consider removing phrasing “trend towards significance” as it not sound in the study of statistics. Consider reporting effect size for stronger analysis to assess the magnitude of difference, if data is available. A minimal suggested revision would be to change the phrasing.

**Do you want your identity to be public for this peer review?** For information about this choice, including consent withdrawal, please see our Privacy Policy

Reviewer #1: No

---

## [Editor Report · Decision Letter 1]

Factors contributing to neonatal mortality in a tertiary center in south West Bank: A Single-Center Retrospective Cohort Study

PGPH-D-25-00456R1

Dear dr khdour,

We are pleased to inform you that your manuscript 'Factors contributing to neonatal mortality in a tertiary center in south West Bank: A Single-Center Retrospective Cohort Study' has been provisionally accepted for publication in PLOS Global Public Health.

Best regards,

Sanghyuk S Shin

Academic Editor